# Obstructive Sleep Apnea and Recovery in Athletes: BMI and Neck Circumference and Their Impact on Recovery Capacity and Injury Risk

**DOI:** 10.3390/life16010076

**Published:** 2026-01-04

**Authors:** Marcin Sikora, Mariusz Panek, Olga Łakomy, Szymon Siatkowski, Emilia Głowacka, Aleksandra Żebrowska

**Affiliations:** 1Institute of Sport Sciences, Academy of Physical Education in Katowice, 72A Mikolowska Street, 40-065 Katowice, Polands.siatkowski@awf.katowice.pl (S.S.); emiliaglowacka123@gmail.com (E.G.); a.zebrowska@awf.katowice.pl (A.Ż.); 2Department of Otorhinolaryngology and Oncological Laryngology, Faculty of Medical Sciences in Zabrze, Medical University of Silesia in Katowice, Marii Curie-Sklodowskiej 10, 41-800 Zabrze, Poland; m.h.panek@gmail.com

**Keywords:** obstructive sleep apnea, recovery capacity, injury risk, athletes, sports nutrition, sleep–recovery interaction, body mass index, neck circumference, body composition, sleep disorders

## Abstract

Obstructive sleep apnea (OSA) is an underdiagnosed condition in athletes, strongly influenced by anthropometric factors such as body mass index (BMI) and neck circumference, which often reflect long-term sport-specific training and nutritional strategies. This review examines the impact of OSA on athletes’ recovery, injury risk, and performance, with emphasis on BMI and neck circumference as key risk markers. A comprehensive search was conducted in PubMed, Web of Science, EBSCO, and Google Scholar for studies published between 2015 and 2025. Eleven studies met the inclusion criteria. The findings indicate a substantial prevalence of OSA in athletes, particularly those involved in collision sports such as rugby and American football, where higher BMI and increased neck circumference were consistently associated with increased risk of OSA. OSA was associated with reduced sleep quality, hormonal disruption, excessive daytime sleepiness, and slower reaction times. These factors may collectively impair recovery, increase injury susceptibility, and negatively affect aerobic capacity, cognitive function, and sport-specific performance. The results highlight the need for routine sleep screening in athletes, especially those with elevated BMI or larger neck circumference. Early detection and management of OSA may improve recovery and performance. Further longitudinal studies are needed to evaluate the long-term effects of OSA treatment in athletic populations.

## 1. Introduction

Sleep is increasingly recognized as a key element in training periodization and overall athletic preparation. For athletes, sleep is particularly important due to the physiological changes induced by training. During sleep, numerous reparative processes occur, including the regeneration of damaged muscle fibers or the resynthesis of energy substrates, especially muscle glycogen. Sleep is also a critical period for the secretion of anabolic hormones such as testosterone and growth hormone, which play essential roles in recovery and performance enhancement in athletes [1]. Importantly, these recovery processes are modulated not only by sleep duration and quality, but also by nutrition-related body composition, particularly body mass index (BMI) and regional fat distribution, which may influence both sleep architecture and subsequent training adaptation.

A range of psychological, behavioral, and environmental factors can impair sleep quality in athletes. Psychological stress, anxiety related to competition, and emotional tension are frequently reported among elite athletes, often resulting in sleep onset insomnia or nighttime awakenings [2,3]. Additionally, high-intensity physical training performed in the late evening hours may raise body temperature and sympathetic nervous system activation, delaying the onset of restorative sleep [4]. Consuming a hyperglycemic diet close to bedtime has also been associated with poor sleep efficiency, increased sleep latency, and more nighttime awakenings [5,6]. Other behavioral factors, such as alcohol or caffeine intake in the evening, may interfere with circadian rhythms and suppress melatonin secretion, impairing sleep onset and quality [7]. Environmental conditions, such as high or low ambient temperature, noise, or light exposure, also play a significant role. For instance, exposure to blue light from electronic devices in the evening can delay melatonin release and alter circadian phase timing [8]. These non-pathological factors are prevalent in athletic settings and may cumulatively contribute to chronic sleep restriction. These factors may further interact with body composition and anthropometric characteristics, potentially modifying the risk of sleep-disordered breathing and exercise recovery disturbances in athletes.

Beyond these factors, obstructive sleep apnea (OSA) emerges as a serious and often underdiagnosed sleep disorder in athletic populations. OSA is characterized by repeated episodes of upper airway obstruction during sleep, resulting in intermittent hypoxia, sleep fragmentation, and sympathetic hyperactivation [9,10]. In athletes, OSA may remain clinically silent but still be associated with reduced deep sleep (non-rapid eye movement sleep, NREM), impaired secretion of anabolic hormones such as growth hormone and testosterone, and prolonged recovery time [11]. Chronic sleep disruption may be associated with impaired physical regeneration, reduced cognitive alertness, and poorer athletic performance. Given its multifactorial impact, OSA deserves focused attention as both a sleep disorder and a limiting factor in post-exercise recovery [12,13,14].

Although the effects of OSA on metabolic health, cardiovascular function, and cognitive performance have been well documented [11,14,15,16], fewer studies have directly examined its impact on athletic recovery. Previous studies have shown that OSA is associated with endocrine disruptions, particularly reduced growth hormone (GH) secretion, which may impair recovery processes [14,17]. In addition, chronic intermittent hypoxia associated with OSA contributes to oxidative stress and systemic inflammation, which further delay recovery processes and may increase the risk of overuse injuries [11,18,19].

The chronic sleep compartmentalization associated with OSA can limit the secretion of growth hormone (GH), which is an important factor in muscle regeneration and tissue recovery [9]. In addition, repeated episodes of hypoxia can lead to oxidative stress and increased inflammation, which can further impair the body’s regenerative capacity [20]. Research involving athletes has demonstrated that increased body mass, higher BMI, and larger neck circumference are important factors associated with the risk of OSA. These anthropometric characteristics are often shaped by long-term sport-specific nutritional strategies and body composition goals, linking sleep-disordered breathing directly with nutrition and training adaptation. Consequently, athletes engaged in strength-based disciplines and contact team sports are considered to be at an increased risk of developing OSA [16]. In these groups, elevated BMI and increased neck circumference often represent functional adaptations to training and nutritional practices, but may simultaneously increase vulnerability to sleep-disordered breathing and impaired post-exercise recovery. Therefore, the question of whether OSA may be a hidden limiting factor in athletes’ recovery and whether its diagnosis and treatment can improve exercise capacity and reduce the risk of injury becomes relevant.

The purpose of this review was to evaluate the impact of OSA on athletes’ recovery, performance, and injury risk, with particular emphasis on the roles of anthropometric factors such as BMI and neck circumference in modulating sleep–recovery interactions. A systematic literature search was conducted in PubMed, Web of Science, EBSCO, and Google Scholar for studies published between 2015 and 2025. Eligible studies included amateur and professional athletes and reported outcomes related to recovery, performance, or risk of injury, assessed either directly (e.g., injury incidence, recovery markers, performance tests) or indirectly (e.g., sleep quality, hormonal profiles, subjective fatigue).

## 2. Materials and Methods

This systematic review was conducted in accordance with the PRISMA 2020 guidelines (Preferred Reporting Items for Systematic Reviews and Meta-Analyses; PRISMA Group, Oxford, UK). A completed PRISMA 2020 checklist is provided in the Appendix A, and the full PRISMA 2020 flow diagram is presented in Figure 1. The methodology included a comprehensive literature search, clear eligibility criteria, and a structured synthesis of the findings. The electronic search was conducted in PubMed/MEDLINE (National Library of Medicine, Bethesda, MD, USA), Web of Science (Clarivate Analytics, Philadelphia, PA, USA), and EBSCO (EBSCO Information Services, Ipswich, MA, USA) as the primary bibliographic databases. In addition, Google Scholar (Google LLC, Mountain View, CA, USA) was used as a supplementary source to identify potentially relevant studies not indexed in traditional databases (e.g., early-access publications and sport science journals with limited indexing). To minimize selection bias, only the first 200 results sorted by relevance were screened manually. The review was conducted in accordance with the Preferred Reporting Items for Systematic Reviews and Meta-Analyses guidelines (PRISMA) [21], ensuring transparency in the study selection process. Verification of the research methodology was carried out using the Critical Appraisal Skills Programme (CASP) tool [22], which made it possible to assess the quality of the included articles. Ethical approval was obtained from the local ethics committee under approval number Nr 5/II/2025, in preparation for planned research involving athletes. This review article lays the groundwork for further ethically approved studies in this field. Although this systematic review was not registered in PROSPERO due to its exploratory and narrative nature, registration is planned for future experimental studies.

### 2.1. Search Strategy

The search process was conducted on 12 May 2025, and included publications published within the last ten years (2015–2025), to identify current research in sports medicine and sleep disorders. The primary search areas comprised three main topic categories: (1) Obstructive sleep apnea and sleep-disordered breathing—terms included “Obstructive Sleep Apnea” and “OSA”. (2) Athletes in various sports—terms included “Athletes”, “Professional Athletes”, and “Elite Athletes”. (3) Recovery and physical performance—terms included the phrases “Recovery” and “Regeneration”. (4) Anthropometric and body composition factors related to OSA (e.g., BMI, neck circumference, body fat).

To optimize the search results, logical AND and OR operators were used to both combine a wide range of search terms and eliminate irrelevant results. The search strategy included both Medical Subject Headings (MeSH) and terms from the free text, ensuring a comprehensive analysis of the literature. Detailed search strategies, including full search strings for each database and the corresponding number of retrieved records, are provided in Appendix A. Briefly, combinations of terms related to obstructive sleep apnea, athletes, recovery or performance outcomes, and anthropometric factors (e.g., BMI, neck circumference, body composition) were applied using Boolean operators. Filters were applied to limit results to articles published in English between 2015 and 2025. Google Scholar was screened as a supplementary source (first 200 results by relevance) to identify potentially missing or early-access records.

### 2.2. Inclusion and Exclusion Criteria

To increase the relevance of this review, the following study selection criteria were used:

Inclusion criteria:Time range: Articles published between 2015 and 2025.Language of publication: English language publications only, to avoid errors in interpretation.Study population: Studies on amateur and professional athletes.Type of research: Systematic reviews, meta-analyses, clinical and observational studies included.OSA diagnostic methods: Polysomnography, polygraphy, STOP-BANG questionnaire (Charité–Universitätsmedizin Berlin, Berlin, Germany), Berlin Questionnaire (Charité–Universitätsmedizin Berlin, Berlin, Germany). The Apnea–Hypopnea Index (AHI), which quantifies the number of apneas and hypopneas per hour of sleep, is commonly used to assess the severity of OSA. An AHI of ≥5 is generally considered diagnostic for mild OSA.Accessibility: Studies available in full text or obtainable through institutional databases were preferred.Anthropometric data: Studies reporting anthropometric parameters such as body mass index (BMI), neck circumference or body composition indicators were prioritized where available.

Exclusion criteria:Population: Studies involving the general population and non-sports-related OSA patients were excluded.Scope: Papers focusing exclusively on psychological aspects of OSA, non-recovery-related pharmacological interventions, or validation of diagnostic tools were not included.Case studies: Case studies of individual athletes with limited statistical value were excluded.Methodological evaluation: Articles rated negatively in CASP in terms of methodological soundness were excluded.

### 2.3. Evaluation of the Quality of the Reviewed Articles

The methodological quality and risk of bias of the included studies were assessed independently by two reviewers using validated tools appropriate to the study design. Randomized controlled trials were evaluated using the Cochrane Risk of Bias 2.0 tool (RoB 2.0; Cochrane Collaboration, London, UK)), which assesses bias across five domains (randomization process, deviations from intended interventions, missing outcome data, measurement of the outcome, and selection of the reported result). Observational cohort and cross-sectional studies were assessed using the Newcastle–Ottawa Scale (NOS Ottawa Hospital Research Institute, Ottawa, ON, Canada), evaluating selection, comparability, and outcome/exposure domains. Qualitative and mixed-methods studies were assessed using the Critical Appraisal Skills Programme (CASP) checklists (CASP, Oxford, UK). For each study, an overall risk-of-bias judgment (low, moderate, or high) was assigned based on predefined criteria. Disagreements between reviewers were resolved through discussion. A detailed study-level summary of the risk-of-bias assessment is provided in Appendix A. A summary of the risk-of-bias assessment is presented in Table 1. Of the eleven included articles, ten were original empirical studies and were therefore subjected to formal risk-of-bias assessment, whereas narrative reviews were not assessed Kölling et al., 2019 [23].

Based on this assessment, publications focusing on validation of diagnostic tools (e.g., [24]), cardiovascular aspects of sleep disorders (e.g., [16]), and analyses of multisystem health problems in former athletes were excluded. Papers focusing on the impact of obesity [25] or simulated altitude conditions [26] were also excluded. Detailed methodological analysis allowed us to reduce the impact of bias in the results and increase the reliability of the conclusions. Of the 11 studies, 7 achieved high methodological quality, while 4 had a moderate risk of error, mainly due to the limited sample, the vast majority of studies’ focus on males, or the lack of a control group. Studies using polysomnography and polygraphy as methods to diagnose OSA were judged to be more reliable compared to studies based on screening questionnaires alone. Additionally, differences in the populations of the athletes studied may have influenced the discrepancies in results between studies. Particular attention was also paid to whether anthropometric characteristics such as BMI and neck circumference were objectively measured or self-reported, as this could influence the validity of OSA risk assessment. Detailed bibliographic data are shown in Table 2.

### 2.4. Data Extraction

Data from the selected publications were independently extracted by two reviewers. In cases of discrepancies in data interpretation, discussions were held to reach consensus, or a third reviewer was consulted. Extracted data included study population, age, BMI, neck circumference (if reported), type of sport, presence of OSA, diagnostic methods used, and key findings related to recovery, performance, and injury risk.

## 3. Results

Eleven studies fulfilling the inclusion criteria were eligible for analysis. Initially, 570 articles were identified in the databases, of which 133 were from EBSCO, 144 from Web of Science, 74 from Google Scholar, and 293 from PubMed/MEDLINE. After applying the filters, the number of potentially relevant publications was reduced to 13, 15, 14, and 74, respectively. After full text analysis and methodological evaluation according to the CASP, Newcastle–Ottawa, and RoB 2.0 tools, 11 studies were qualified for the final analysis. The full selection process is presented in the PRISMA 2020 flow diagram (Figure 1).

The total number of participants in all analyzed studies was 1034, comprising 711 males and 323 females, representing different sports. The most frequently studied groups were players of team sports, such as rugby and American football, in which OSA occurred with high frequency. The age of the subjects ranged from 18 to 45 years and BMI values ranged from 22 to 36 kg/m^2^, with higher values predominantly reported in contact sport athletes. Where available, increased neck circumference was also frequently observed in these groups and was often associated with a higher prevalence of OSA. Detailed demographic data are shown in Table 3. It is worth noting that the analyzed reports did not use objective monitoring of physical activity, which significantly reduces the value of the results obtained.

The literature analysis examined reviewed studies on obstructive sleep apnea (OSA) and its impact on recovery capacity and injury risk in athletes. The data presented in Table 3 are from studies including both team and individual athletes, representing a range of performance levels from collegiate to professional athletes.

The results of the studies analyzed indicate a high prevalence of OSA among athletes, particularly in contact sports such as rugby and American football. Suzuki et al. [13] revealed that up to 64.3% of rugby players had symptoms of OSA, a finding that was also confirmed by studies by Caia et al. [18] and Dunican et al. [27], which indicated a higher prevalence of the condition among players playing in forwards positions. In American football, as shown by Peck et al. [16] and Dobrosielski et al. [19], offensive and defensive line players were particularly prone to OSA, which was associated with higher BMI and larger neck circumference. OSA symptoms were also observed in swimmers [28] and alpine climbers [29], albeit at a lower severity.

All articles analyzed were grouped according to the observations made by the authors, who described the effects of OSA on recovery, injury risk, and physical performance.

Effects of OSA on athlete recovery

Sleep plays a fundamental role in athletic recovery by supporting tissue repair, hormonal balance, and neuromuscular regeneration. During deep non-REM sleep stages, particularly stages 3 and 4, the body secretes growth hormone (GH), stimulates protein synthesis, replenishes glycogen stores, and modulates immune function. This phase of sleep is also essential for removing metabolic waste from the brain and reducing inflammation. Disruptions to sleep architecture—such as those caused by obstructive sleep apnea—can impair these processes and lead to incomplete recovery and overtraining symptoms [1,4,12]. The studies analyzed indicate that OSA negatively affects athletes’ recovery. Suppiah et al. [30] found that 45.2% of young athletes experienced poor sleep quality, which was particularly evident in team sports. Similarly, Dunican et al. [27] found that 24% of union rugby players had OSA, which was associated with higher BMI and neck circumference. Furthermore, elevated BMI and increased neck circumference may aggravate sleep fragmentation and nocturnal hypoxia, thereby further compromising post-exercise recovery capacity. In contrast, Swinbourne et al. [31] reported that 50% of the athletes surveyed reported sleep problems and 28% experienced excessive daytime sleepiness. Surda et al. [28] found that 30% of elite swimmers had symptoms of OSA, with morning training further compromising sleep quality. In professional rugby league athletes, BMI showed a moderate positive correlation with AHI (r = 0.38, *p* = 0.04), indicating that increasing body mass index may contribute to more severe sleep-disordered breathing [18]. Moreover, skinfold thickness, used as a proxy for body fat accumulation, was moderately correlated with AHI (r = 0.40, *p* = 0.05), suggesting a potential role of increased adiposity in worsening OSA severity among athletes [18]. Peck et al. [15] found that football linemen had a higher rate of OSA than athletes, which was linked to higher body weight and neck circumference. Research on alpine climbers by Ortiz-Naretto et al. [29] also indicated that athletes with OSA had difficulty adapting to hypoxia, shown to negatively affect recovery. Nabhan et al. [32] also suggest an effect of OSA on the deterioration of recovery, which has been linked to depression and anxiety in Olympic and Paralympic athletes. Suzuki et al. [13] evaluated the impact of MAD therapy in rugby players and found improvements in sleep quality, recovery process, and reaction time following treatment. Mandibular advancement devices (MADs) are oral appliances designed to reposition the lower jaw forward during sleep. This mechanical adjustment helps to keep the upper airway open, reducing the occurrence of obstructive events such as apneas and hypopneas.

Effect of OSA on injury risk

Several studies reported associations between OSA and injury-related markers in athletes. Dobrosielski et al. [19] observed that collegiate football players with OSA exhibited slower reaction times and reduced alertness. Similarly, Peck et al. [15] reported a higher injury rate among football linemen classified as having OSA compared to other athletes. In professional rugby players, OSA was reported predominantly among forwards, who also demonstrated significantly greater neck circumference compared to backs (42.7 ± 3.1 cm vs. 39.0 ± 4.1 cm, ES = 1.04, *p* = 0.05) [18]. Dunican et al. [27] further noted that rugby players with OSA more frequently reported excessive daytime sleepiness. In an intervention study, Suzuki et al. [13] reported improvements in reaction time following mandibular advancement device therapy.

Effects of OSA on performance

Several studies reported associations between OSA, sleep disruption, and reduced exercise capacity in athletes. Swinbourne et al. [31] and Suppiah et al. [32] observed that athletes with sleep deficits associated with OSA demonstrated impaired performance outcomes. Similarly, Dunican et al. [27] and Caia et al. [18] reported poorer performance in exercise tests among rugby players with OSA. In swimmers, irregular sleep patterns and OSA-related disturbances were associated with reduced ability to perform high-intensity aerobic efforts [28]. Ortiz-Naretto et al. [29] further noted that mountaineers with OSA exhibited metabolic limitations and reduced exercise capacity under hypoxic conditions

Prevalence of OSA in different groups of athletes

The reported prevalence of OSA varied across sports and athlete populations. Suzuki et al. [13] reported that 64.3% of rugby players exhibited symptoms consistent with OSA. Similarly, Caia et al. [18] and Dunican et al. [27] reported OSA prevalence in professional rugby players, with higher occurrence observed among forwards compared to backs. In American football players, Peck et al. [15] and Dobrosielski et al. [19] reported a higher prevalence of OSA risk, particularly among players with larger body mass profiles. Surda et al. [28] and Ortiz-Naretto et al. [29] further reported the presence of OSA-related disturbances in swimmers and climbers, although generally at lower severity compared to collision sports.

## 4. Discussion

The main achievement of this review is a comprehensive analysis of the impact of obstructive sleep apnea (OSA) on recovery capacity, physical performance, and injury risk among athletes from different sports. While previous studies—such as the International Olympic Committee consensus [33] or the work of Kölling et al. [34] and Fullagar et al. [4]—have focused on the general role of sleep in optimizing the health and performance of athletes, this review is the first to systematically examine a specific sleep disorder—namely, OSA—and its potential impacts on athlete functioning. In addition, our synthesis emphasizes the role of key anthropometric factors, particularly body mass index (BMI) and neck circumference, which bridge sports nutrition, body composition, and sleep-disordered breathing in the context of post-exercise recovery.

The novelty of this review lies in the combination of data from a wide range of athletes—from contact and strength-based athletes to endurance athletes, alpine mountaineers, and Paralympic athletes—to capture both the scale of the problem and its specific determinants. Unlike previous studies, this one not only identifies the problem but also considers the available data on potential interventions, such as the use of mandibular advancement devices (MADs) which, although described to date in isolated studies, may provide a practical alternative to standard treatment for athletes with OSA [13]. Moreover, by systematically extracting information on BMI, neck circumference, and (where available) body composition, this review highlights anthropometric profiles that may be particularly vulnerable to OSA-related disturbances in sleep and recovery.

Prevalence of OSA in different athletes

The evidence collected clearly shows that the prevalence of OSA among athletes is not limited to one category of sport or athlete phenotype. Traditionally, the greatest focus has been on contact sports athletes such as American football and rugby, where athletes are characterized by high BMI and large neck circumference—anthropometric features considered major risk factors for OSA [34] and largely shaped by long-term training and nutrition-related body composition strategies. The studies by Dunican et al. [27], Peck et al. [15], and Suzuki et al. [13] confirm a high prevalence of OSA ranging from 24% to over 60% among rugby and football players, with apnea prevalence significantly correlated with higher BMI, larger neck circumference, and, in some cohorts, increasing age. However, equally interesting, and less intuitive, are the findings for athletes from seemingly less vulnerable groups. Surda et al. [28] demonstrated that elite swimmers, despite a relatively low BMI, present a significant prevalence of OSA symptoms. The authors suggest that this may be related to chronic chlorine exposure leading to non-allergic rhinitis and mouth breathing during training. This finding confirms that classical risk criteria for OSA may be insufficient in the assessment of athletes and should be supplemented by environmental factors and training specificity. Nabhan et al. [32], on the other hand, highlight the population of Paralympic athletes, a group that has to date been overlooked in the context of sleep research. In a study of nearly 1000 athletes, they found that those with disabilities experienced sleep problems significantly more often than their non-disabled counterparts. This may be due not only to anatomical factors but also to mobility limitations, chronic pain, or neurophysiological changes associated with the underlying disease. High-altitude mountaineering is also an interesting phenomenon. Ortiz-Naretto et al. [29] described the effect of OSA on hypoxia tolerance in high-altitude conditions, highlighting the synergistic effects of hypoventilation and reduced oxygen partial pressure. Their observations suggest that even minor respiratory disturbances can lead to significant impairment of performance and increased risk of health complications under these conditions. Across the reviewed studies, a consistent pattern emerged in which elevated BMI and increased neck circumference were more frequently observed among athletes classified as being at higher risk for OSA, particularly in collision and strength-based sports. These anthropometric characteristics were repeatedly reported in rugby and American football players, whereas athletes from endurance-based disciplines and aquatic sports generally presented lower BMI but were not free from OSA-related disturbances. This indicates that, while body mass and neck circumference are important markers of OSA risk in athletes, they do not fully explain the occurrence of sleep-disordered breathing across different sporting environments.

Therefore, the data from this review show that OSA is a cross-sectional problem, present in a diverse range of sporting environments, and should not be treated as the domain of weight-bearing athletes alone. The variety of risk factors—from anatomical (e.g., BMI, neck circumference, craniofacial structure) to environmental and functional—calls for a more individualized diagnostic approach and greater clinical vigilance among professionals working with athletes [4,13,15,28,29,31,33,34,35].

Impact of OSA on recovery, performance, and injury risk

OSA, which is a chronic disorder, affects the athlete’s body in a multidimensional way, disrupting not only sleep but also broader physiological functioning. One key area is post-workout recovery. The hypoxia and sleep fragmentation associated with OSA have been linked to altered secretion of growth hormone and testosterone [17,18,30], key to repair and adaptation processes [18,30]. These disturbances may be particularly pronounced in athletes with higher BMI and greater adiposity, where excess body mass and increased neck circumference not only elevate the risk of OSA but may also exacerbate nocturnal hypoxia and sleep fragmentation, further compromising recovery capacity. Such a condition may be associated with poorer recovery of muscle microdamage, increased post-workout soreness and longer psychophysical recovery times [35,36]. Furthermore, OSA has been shown in studies to be associated with chronic activation of the immune system, including increased levels of pro-inflammatory cytokines such as interleukin-6 (IL-6) and tumor necrosis factor alpha (TNF-α) [11]. Elevated levels of these inflammatory mediators may further delay regenerative processes in muscle tissue and disrupt metabolic homeostasis [11]. Chronic inflammation, in combination with hypoxia and sleep fragmentation, can negatively affect athletes’ recovery from exercise and increase susceptibility to overload and micro-injury, particularly in those with higher BMI and unfavorable body composition profiles [37].

Additionally, repeated episodes of hypoxia negatively affect aerobic capacity, which is particularly dangerous in endurance sports and those requiring high aerobic fitness. Ortiz-Naretto et al. [29] showed that athletes at high altitude with unrecognized OSA presented greater adaptation difficulties and significant decreases in nocturnal saturation. Furthermore, Dunican et al. [27] and Suzuki et al. [13] showed athletes with OSA were characterized by reduced cognitive concentration and vigilance, impaired reaction time, and higher levels of daytime fatigue, primarily associated with excessive sleepiness and reduced alertness. In several of the analyzed studies, athletes with OSA also presented higher BMI or greater neck circumference, suggesting that unfavorable anthropometric profiles may intensify the physiological burden of nocturnal hypoxia. This interaction between body composition, sleep-disordered breathing, and oxygen transport is particularly relevant in sports where both high body mass and high aerobic capacity are required, and should be considered when planning nutrition and recovery strategies.

It is also worth noting that OSA has a significant impact on injury risk. Swinbourne et al. [31] demonstrated that athletes with sleep deprivation are at a higher risk of soft tissue injury, which may be a result of reduced recovery, but also slowed motor response and impaired attention. The sleep deprivation and hypoxemic episodes characteristic of OSA negatively affect neurocognitive functions, including reaction time, concentration, and the ability to make quick decisions, which significantly increases the risk of injury in high-intensity physical activity [38,39,40]. In addition, chronic sleep fragmentation—typical of OSA—is associated with repeated micro-awakenings that disrupt the structure of deep sleep (NREM3) responsible for muscle recovery and cognitive function [1]. Research indicates that athletes with higher levels of sleep fragmentation have been associated with a higher risk of injury, impaired balance, and delayed psychomotor response [1,39]. Milewski et al. [41] found that athletes who slept under 8 h per day exhibited a significantly increased incidence of musculoskeletal injuries almost twice as often as their better-rested peers. Taken together, the available evidence suggests that OSA is associated with injury-related risk factors in athletes, particularly through impaired recovery, reduced alertness, and delayed reaction time. However, due to the observational nature of most studies, a direct causal relationship between OSA and injury incidence cannot yet be established.

From a sports nutrition perspective, many athletes—particularly those in contact and strength-based disciplines—intentionally increase their body mass, muscle mass and, indirectly, neck circumference to enhance performance. While such adaptations may be beneficial for force production and physical dominance, they may inadvertently increase the risk of OSA and exacerbate sleep-related disturbances in recovery. Our findings therefore support an integrated approach in which nutrition planning, body composition targets, and sleep health are considered together, rather than in isolation. In practice, this means that anthropometric changes such as rises in BMI or neck circumference should trigger not only performance-related discussions but also a re-evaluation of sleep quality, OSA risk, and recovery strategies. Overall, the available evidence indicates that the effects of OSA in athletes cluster around three main domains—impaired recovery processes, reduced physical and cognitive performance, and an unfavorable injury risk profile—although the strength of evidence varies across these outcomes.

Intervention strategies and practical recommendations

Both the diagnosis and treatment of OSA should be adapted to the specific context of athletic environments. Routine screening in athletes should therefore combine validated sleep questionnaires with simple anthropometric assessments, including BMI and neck circumference, which can be easily monitored within existing sports nutrition and medical support structures. Continuous positive airway pressure (CPAP) remains the gold standard therapy for OSA in the general population [42]. This method is considered highly effective, with studies showing that regular use of CPAP improves sleep quality, reduces daytime sleepiness, and enhances cognitive function and overall quality of life [43]. Although the implementation of CPAP therapy in athletes may pose specific challenges—such as limited tolerance for wearing a mask during sleep—it should not be disregarded [11]. An inadequately fitted CPAP mask may also cause discomfort, air leakage, and reduced adherence to therapy. Individualized equipment fitting, combined with targeted education regarding the mechanisms of CPAP therapy and its physiological benefits for athletes, may significantly improve adherence to therapeutic recommendations within this population [13]. An alternative treatment option for CPAP therapy for selected athletes with OSA is the use of mandibular advancement devices. Suzuki et al. [13] demonstrated that MAD therapy can improve sleep quality, cognitive function, and reaction time in athletes with OSA symptoms [13]. In addition to medical interventions, non-pharmacological strategies play a critical role in supporting sleep health among athletes. Educational programs promoting sleep hygiene, including awareness of sleep’s role in recovery and personalized counseling, have been shown to improve both sleep duration and quality in elite sports settings [44]. Maintaining a consistent circadian rhythm, regardless of training schedules, has also been linked to better sleep outcomes and reduced sleep disturbances [4]. Limiting exposure to blue light in the evening, particularly from screens, has been shown to decrease sleep onset latency and increase sleep depth [8]. Furthermore, scheduling training sessions in alignment with athletes’ natural chronotypes can support more efficient recovery and better circadian alignment [8].

From a sports medicine perspective, incorporating OSA screening into routine medical evaluations may be particularly beneficial in high-risk sports such as rugby, American football, and endurance disciplines. Regular sleep assessments, greater awareness of OSA symptoms among coaching and medical staff, and timely diagnosis may not only improve athletes’ health and well-being but also enhance performance and reduce the risk of injury.

Limitations and future research directions

Despite the growing number of studies on sleep in sport, there is still a lack of high-quality data on OSA in this population. Most available studies rely on screening tools such as the Epworth Sleepiness Scale (ESS) or the Pittsburgh Sleep Quality Index (PSQI), whereas a full diagnosis of OSA requires polysomnography or at least polygraphy. Another important limitation of the available evidence is the strong predominance of male athletes across the included studies. This limits the generalizability of the findings to female athletic populations, who may differ in OSA risk profiles due to sex-specific anatomical, hormonal, and physiological factors. Given the increasing participation of women in elite sport, future studies should deliberately include female athletes to better characterize the prevalence, risk factors, and recovery-related consequences of OSA across sexes. Another important methodological limitation is the absence of randomized controlled trials (RCTs) among the included studies. Most of the available evidence is derived from cross-sectional or observational designs, which restricts the ability to determine whether OSA directly contributes to impaired recovery, reduced performance, or increased injury risk in athletes, or whether these associations are influenced by confounding factors such as training load, body composition, or sport-specific demands. Future research should prioritize well-designed RCTs evaluating both diagnostic and therapeutic interventions for OSA in athletic populations to strengthen causal inference and inform evidence-based clinical decision-making. A further limitation of the current evidence base is the lack of long-term follow-up data evaluating the effects of OSA treatment in athletes. Although short-term improvements in sleep quality, cognitive function, reaction time, and subjective recovery have been reported following interventions such as mandibular advancement device therapy, the durability of these effects across prolonged training cycles, competitive seasons, or athletic careers remains unclear. Moreover, there is limited information on whether OSA treatment leads to sustained reductions in injury incidence or improvements in cardiometabolic health markers in athletic populations. Longitudinal studies with extended follow-up periods are therefore needed to determine the long-term effectiveness and practical relevance of OSA treatment strategies in sport. Finally, differences in clinical definitions of OSA and apnea–hypopnea index (AHI) values make it difficult to compare results between studies. The AHI (i.e., the number of apneas and shortness of breath per hour of sleep) can be calculated using different methods, depending on the diagnostic criteria adopted—e.g., the degree of desaturation required or the presence of micro-awakenings. In addition, the diagnostic thresholds used (e.g., AHI ≥5, ≥10, or ≥15) and the test methods used (polysomnography vs. screening) significantly affect the classification of OSA severity and make direct comparisons between tests difficult. Moreover, one of the most important methodological limitations of the available literature is the complete lack of objective assessment of daily physical activity and its correlation with sleep and wakefulness measurements using wearable technologies such as accelerometers or multisensor devices. None of the studies analyzed included accelerometer-based monitoring to quantitatively determine daily training load, spontaneous physical activity, or actual sleep patterns, even though these factors directly affect recovery processes and may influence the severity of OSA. This is particularly relevant given the variable training conditions characteristic of the athlete population. This gap limits the interpretation of the OSA-related findings, as sleep-disordered breathing cannot be adequately contextualized in relation to an athlete’s actual training load, recovery requirements, and circadian stressors. Future studies should incorporate wearable monitoring devices—combining accelerometry, heart rate or heart rate variability, and nocturnal oxygen saturation—to enable long-term ecological assessment of sleep, recovery, and OSA-related disturbances under real-world training and competition conditions.

## 5. Conclusions

Obstructive sleep apnea (OSA) is a significant but still underestimated problem in the sporting environment. The results of this review show that the disorder is prevalent not only among athletes with a high BMI and large neck circumference, such as American football and rugby players, but also in seemingly low-risk groups, including swimmers, Paralympians, and alpine climbers. It is associated with poorer recovery quality, reduced aerobic capacity, and impaired cognitive function, and may be linked to a higher risk of injury, particularly in athletes with elevated BMI and unfavorable anthropometric profiles; however, the predominantly observational evidence does not allow causal relationships to be established. Despite this, screening and diagnosis of OSA remain rare in standard care for athletes, and are mostly questionnaire-based, which significantly limits diagnostic accuracy and makes it difficult to identify cases requiring treatment. Furthermore, simple anthropometric markers such as BMI and neck circumference are rarely used systematically in athletic screening, despite their practical usefulness in identifying high-risk individuals. Alternatives, such as mandibular advancement devices (MADs), show promise in improving sleep quality and cognitive function, as shown in studies with athletes. Equally important is the role of education and prevention, in terms of sleep hygiene, recognizing symptoms suggestive of sleep-disordered breathing, and understanding how long-term nutritional strategies and body composition changes may influence OSA risk. Sports physicians, coaches, and the athletes themselves should be aware of the importance of sleep and the potential consequences of sleep disorders.

Therefore, it is recommended to introduce routine screening for OSA among athletes, especially in high-risk sports, incorporating both sleep questionnaires and basic anthropometric assessment (BMI and neck circumference). Further research, particularly long-term and including female populations, is also needed to better understand the impact of OSA treatment on the long-term health and performance of athletes. Recognizing, preventing, and effectively treating OSA can not only improve athletes’ quality of life but also provide a real advantage in terms of sports performance, optimization of recovery capacity, and minimizing the risk of injury, particularly when integrated with personalized nutritional and training strategies. Where feasible, incorporating objective wearable-based monitoring may further enhance screening sensitivity in real-world athletic settings.

## Figures and Tables

**Figure 1 life-16-00076-f001:**
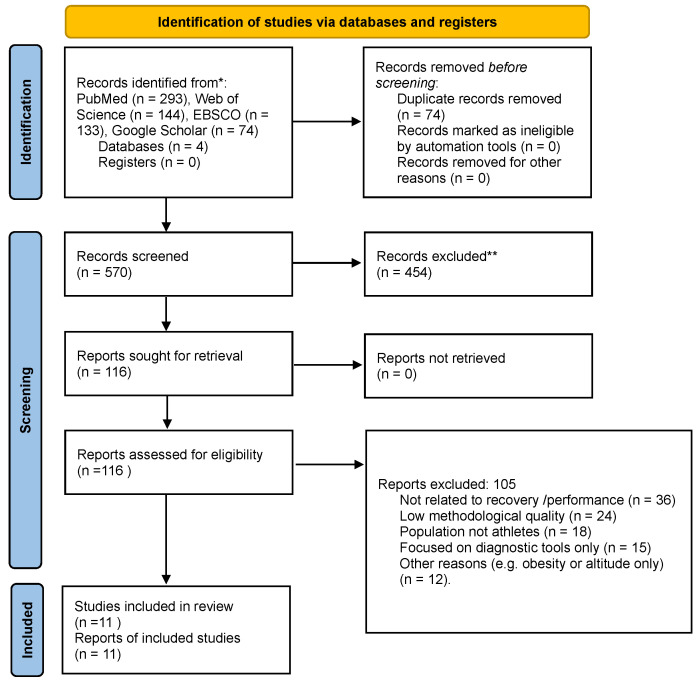
PRISMA 2020 flow diagram illustrating the study selection process. * Records were identified through electronic database searches (PubMed, Web of Science, EBSCO, and Google Scholar). ** Records were excluded after title and abstract screening according to predefined eligibility criteria.

**Table 1 life-16-00076-t001:** Risk-of-bias assessment of included studies.

Study (First Author, Year)	Tool Used	Study Design	Risk of Bias (Overall)	Key Comments
**Suzuki et al., 2022**	CASP	Intervention (MAD)	Low	Well-designed intervention, small sample
**Peck et al., 2019**	RoB 2.0	Observational (comparative)	Moderate	No randomization, different comparison groups
**Dunican et al., 2019**	CASP	Observational (PSG)	Low	Lab PSG, solid control and diagnostics
**Surda et al., 2019**	CASP	Observational	Low	Objective measures, good design
**Ortiz-Naretto et al., 2020**	Newcastle–Ottawa	Observational (altitude)	Moderate	Extreme setting, low sample size
**Caia et al., 2020**	Newcastle–Ottawa	Observational	Moderate	Home PSG, positional variability
**Swinbourne et al., 2016**	CASP	Cross-sectional	Moderate	Subjective data, adequate sample
**Nabhan et al., 2020**	CASP	Retrospective (survey)	Low	Large sample, standardized tool
**Kölling et al., 2019**	N/A	Narrative review	N/A	Narrative review—no bias rating
**Dobrosielski et al., 2016**	RoB 2.0	Observational	Moderate	Subjective risk stratification, limited method
**Suppiah et al., 2021**	CASP	Observational (survey)	Moderate	Self-report PSQI, no objective data

**Table 2 life-16-00076-t002:** Bibliographic data of articles qualified for analysis.

L.p.	Authors, (Year)	Tittle	DOI	Link to Article
**1**	Dunican et al. (2019)	Prevalence of sleep disorders and sleep problems in an elite super rugby union team.	10.1080/02640414.2018.1537092	https://doi.org/10.1080/02640414.2018.1537092
**2**	Swinbourne et al. (2016)	Prevalence of poor sleep quality, sleepiness and obstructive sleep apnoea risk factors in athletes	10.1080/17461391.2015.1120781	https://doi.org/10.1080/17461391.2015.1120781
**3**	Peck et al. (2019)	Examination of Risk for Sleep-Disordered Breathing Among College Football Players	10.1123/jsr.2017-0127	https://doi.org/10.1123/jsr.2017-0127
**4**	Dobrosielski et al. (2016)	Estimating the Prevalence of Sleep-Disordered Breathing Among Collegiate Football Players	10.4187/respcare.04520	https://doi.org/10.4187/respcare.04520
**5**	Caia et al. (2020)	Obstructive sleep apnea in professional rugby league athletes: An exploratory study	10.1016/j.jsams.2020.04.014	https://doi.org/10.1016/j.jsams.2020.04.014
**6**	Suzuki et al. (2022)	Mandibular Advancement Device Therapy in Japanese Rugby Athletes with Poor Sleep Quality and Obstructive Sleep Apnea	10.3390/life12010032	https://doi.org/10.3390/life12010032
**7**	Suppiah et al. (2021)	Sleep Characteristics of Elite Youth Athletes: A Clustering Approach to Optimize Sleep Support Strategies	10.1123/ijspp.2020-0675	https://doi.org/10.1123/ijspp.2020-0675
**8**	Surda et al. (2019)	Sleep in elite swimmers: prevalence of sleepiness, obstructive sleep apnoea and poor sleep quality	10.1136/bmjsem-2019-000566	https://doi.org/10.1136/bmjsem-2019-000566
**9**	Ortiz-Naretto et al. (2020)	Effect of mild obstructive sleep apnea in mountaineers during the climb to Mount Aconcagua	10.5935/1984-0063.20200019	https://doi.org/10.5935/1984-0063.20200019
**10**	Kölling et al. (2019)	Sleep-Related Issues for Recovery and Performance in Athletes	10.1123/ijspp.2017-0746	https://doi.org/10.1123/ijspp.2017-0746
**11**	Nabhan et al. (2020)	Expanding the screening toolbox to promote athlete health: how the US Olympic & Paralympic Committee screened for health problems in 940 elite athletes	10.1136/bjsports-2020-102756	

**Table 3 life-16-00076-t003:** Characteristics and key findings of studies on OSA in athletes.

Author (Year)	Study Characteristics	Key Findings	Key Observations
**Caia et al. (2020)**	22 male professional rugby league athletes; age 23.8 ± 3.6 years; BMI: 30.0 ± 2.2 kg/m^2^; neck circumference: 41.1 ± 4.0 cm; sum of 8 skinfolds: 74.3 ± 17.0 mm; positional groups: forwards (n = 13) vs. backs (n = 9); home-based polysomnography used to diagnose OSA.	OSA was diagnosed in 45% of subjects. Rugby players with OSA were more likely to report poorer recovery-related outcomes between matches.	OSA is common in professional rugby players, with a prevalence comparable to other contact sports. Ethnic differences in OSA prevalence were observed, with players of Polynesian descent more likely to have OSA. Higher BMI and neck circumference were associated with increased AHI. No significant correlation was found between neck circumference and AHI.
**Dobrosielski et al. (2016)**	56 male Division I collegiate American football players; age 19–23 years; BMI: 33.0 ± 5.4 vs. 27.6 ± 3.6 kg/m^2^ (high- vs. low-risk); neck circumference: 44.6 ± 2.2 vs. 41.4 ± 2.8 cm; body composition assessed using DXA (total fat mass, trunk fat mass, abdominal visceral fat).	Sleep-disordered breathing (SDB) was identified in approximately 8% of collegiate American football players. Athletes classified as high-risk for SDB reported poorer sleep quality and higher levels of daytime sleepiness. Reduced aerobic capacity was reported in football players classified as high-risk for SDB.	Higher BMI and greater neck circumference were more frequently observed among athletes classified as high-risk for SDB. Athletes with SDB exhibited higher total and central fat mass, although differences were not statistically significant. Mean total sleep duration was 4.2 ± 1.1 h, and 35% of athletes reported clinically significant daytime sleepiness (ESS ≥ 10).
**Dunican et al. (2019)**	25 male elite rugby union players; age 25 ± 4 years; BMI: 30 ± 3 kg/m^2^ (forwards: 31 ± 3; backs: 29 ± 2); neck circumference: 43 ± 4 cm (above OSA risk threshold); full-night in-laboratory polysomnography; OSA prevalence: 24%.	OSA was identified in 24% of elite rugby players. Players with OSA reported higher levels of daytime sleepiness and fatigue, as well as poorer sleep-related recovery outcomes. Short sleep duration was observed across the cohort, with all players reporting excessive daytime sleepiness (ESS ≥ 10).	No significant associations were observed between BMI, neck circumference, or playing position and apnea–hypopnea index (AHI), despite higher BMI values among forwards. Questionnaire-based screening tools were reported to be ineffective in identifying OSA, whereas polysomnography remained the reference diagnostic method. Reduced oxygen-related exercise tolerance was reported in players with OSA.
**Kölling et al. (2019)**	Narrative review of sleep in elite athletes synthesizing evidence from team and individual sports; focus on sleep quantity, sleep quality, recovery, and performance; discusses prevalence of poor sleep and self-reported symptoms suggestive of sleep-disordered breathing; no original anthropometric or body composition data reported.	The reviewed literature indicated that insufficient sleep was associated with impaired recovery-related processes, altered immune function, and changes in cognitive performance. Sleep deprivation was reported to affect coordination, reaction time, and decision-making, although strength and endurance were not consistently reduced.	Poor sleep quality was reported in approximately 50% of elite athletes, characterized by low sleep efficiency and frequent awakenings. Daytime sleepiness was frequently reported and associated with impaired performance-related outcomes. Self-reported snoring (38%) and apnea episodes (8%) were noted as potential indicators of undiagnosed OSA. External factors such as training schedules, competition timing, travel across time zones, pre-competition stress, and evening exposure to electronic devices were identified as common contributors to sleep disruption.
**Nabhan et al. (2020)**	940 elite athletes (683 Olympians, 257 Paralympians); 462 males and 478 females; representing 36 Olympic and Paralympic sports; OSA risk and sleep quality assessed using the Berlin Questionnaire and the Pittsburgh Sleep Quality Index (PSQI); no objective anthropometric measures (BMI, neck circumference, or body composition) reported.	Poor sleep quality was identified in more than 20% of the athletes. A higher prevalence of OSA risk was reported among Paralympic athletes compared with Olympians (8.6% vs. 3.5%). Paralympic athletes classified as high-risk for OSA reported higher levels of fatigue and poorer recovery-related outcomes.	Females more frequently reported poor sleep quality than males (28.3% vs. 22.2%), while no significant sex differences were observed for OSA risk, anxiety, or depression. Daytime sleepiness and poor sleep quality were associated with higher prevalence of psychological symptoms, including anxiety and depression. Screening questionnaires identified OSA risk in a limited proportion of athletes, highlighting potential underdetection in elite sport populations.
**Ortiz-Naretto et al. (2020)**	8 amateur mountaineers (4 females; mean age 36 years) assessed during a high-altitude expedition; respiratory polygraphy used to identify mild asymptomatic OSA; baseline BMI was higher in the OSA group (27.5 vs. 22.3 kg/m^2^); no data on neck circumference or body composition reported.	Participants with OSA exhibited lower nocturnal oxygen saturation and greater difficulty adapting to hypoxic conditions during ascent. A greater decline in oxygen saturation was observed with increasing altitude in the OSA group compared with controls (SpO_2_: 80% at 746 m vs. 52.5% at 4900 m).	Altitude exposure was associated with an increased number of central apneas and hypopneas in all participants, while obstructive events did not increase. All participants developed high-altitude periodic breathing, with a higher frequency of central apneas observed in individuals with OSA. Participants with OSA more frequently exhibited elevated systolic blood pressure at higher altitudes and showed less favorable physiological adaptation, including higher post-expedition BMI and metabolic markers. None of the participants with OSA reached the summit, and cases of severe altitude sickness required medical treatment.
**Peck et al. (2019)**	21 male Division I American football linemen and 19 track athletes (age 18–22 years); sleep-disordered breathing (SDB) risk assessed using the MAP Index and Epworth Sleepiness Scale (ESS); anthropometry and body composition assessed using DXA; football linemen exhibited higher BMI (35.87 ± 4.77 kg/m^2^), neck circumference (44.36 ± 2.89 cm), and body fat percentage (29.2%) compared with track athletes.	American football players demonstrated a higher SDB risk index (MAP) and a higher prevalence of OSA risk compared with non-football athletes. Football players classified as high-risk for SDB reported greater post-training fatigue and earlier onset of fatigue during match play.	Anthropometric and adiposity-related variables differed substantially between football players and controls (BMI: 35.87 vs. 23.07 kg/m^2^; neck circumference: 44.36 vs. 36.92 cm; body fat: 29.2% vs. 13.1%). Neck circumference, BMI, and body fat percentage were correlated with SDB risk indices. Football players exhibited higher Mallampati scores, longer sleep latency, and greater visceral fat accumulation. Despite elevated SDB risk, ESS scores did not differ between groups, suggesting potential under-recognition of sleep-related symptoms in this population.
**Suppiah et al. (2021)**	135 elite national youth athletes (male and female); mean age 15.5 ± 2.0 years; participants from individual and team sports; mean BMI: 20.9 ± 2.7 kg/m^2^ (range approximately 19.7–25.1 depending on sport); OSA-related symptoms and sleep quality assessed using questionnaires; neck circumference and body composition not reported.	Poor sleep quality was reported in 45.2% of participants (PSQI > 5). Athletes reporting sleep deficiency or OSA-related symptoms more frequently reported fatigue and reduced recovery-related outcomes.	Team-sport athletes exhibited poorer sleep quality than individual-sport athletes, characterized by later bedtimes, lower sleep efficiency, and a higher prevalence of snoring (38.7% vs. 28.8%). No significant associations were observed between BMI and snoring or between snoring and overall sleep quality. Daytime naps were commonly used to compensate for sleep deficits. Caffeine use was identified as a potential contributor to impaired sleep quality, particularly in youth athletes.
**Surda et al. (2019)**	101 elite swimmers (with additional comparison groups including non-elite swimmers, non-swimming athletes, and controls); mean age approximately 18–21 years; mean BMI in elite swimmers: 22.8 ± 3.1 kg/m^2^; OSA assessed using overnight pulse oximetry (oxygen desaturation index, ODI ≥ 5 events/h); actigraphy and pulse oximetry conducted in a subgroup (n = 20); neck circumference and body composition not reported.	OSA was identified in approximately 30% of elite swimmers (ODI ≥ 5), representing a higher prevalence compared with young adult control populations. Elite swimmers reported shorter sleep duration on training days, and shorter sleep duration was associated with reduced aerobic exercise tolerance.	OSA prevalence in swimmers was independent of BMI, with no significant BMI differences between swimmers with and without OSA. Daytime sleepiness was more frequently reported among swimmers and other athletes than in controls. Screening with the Epworth Sleepiness Scale demonstrated limited sensitivity, whereas objective assessment using overnight pulse oximetry identified a greater number of OSA cases. Sleep duration was consistently shorter on training days compared with rest days, particularly before early-morning training sessions. Non-allergic rhinitis was identified as a potential contributor to sleep-related disturbances.
**Suzuki et al. (2022)**	42 male professional Japanese rugby union athletes; mean age 26.3 ± 3.7 years; BMI: 26.7 ± 3.2 kg/m^2^; neck circumference: 41.5 ± 2.1 cm; OSA assessed using a level III portable sleep test (respiratory event index, REI); subgroup of six athletes underwent mandibular advancement device (MAD) therapy.	OSA was identified in 64.3% of professional rugby players. Poor sleep quality (PSQI > 5.5) was reported in 69% of athletes, excessive daytime sleepiness (ESS > 10) in 35.7%, and mean sleep duration was 6.5 h. Following MAD therapy, improvements in REI, ESS scores, sleep quality, and selected cognitive performance measures were reported in the treated subgroup.	Athletes with moderate to severe OSA exhibited larger neck circumference and higher BMI compared with those without OSA. Neck circumference ≥ 42 cm, higher BMI, and increasing age were more frequently observed among athletes classified as high-risk for OSA. Improvements observed following MAD therapy were based on a small intervention subgroup and short-term follow-up.
**Swinbourne et al. (2016)**	175 highly trained male and female team-sport athletes (rugby union, rugby sevens, cricket); age ≥ 18 years; sleep quality, daytime sleepiness, and OSA risk factors assessed using the Pittsburgh Sleep Quality Index (PSQI), Epworth Sleepiness Scale (ESS), and OSA risk questionnaires; no objective anthropometric measures (BMI, neck circumference, or body composition) reported.	Poor sleep quality (PSQI > 5) was reported in approximately 50% of athletes, with mean sleep duration of 7.9 ± 1.3 h. Daytime sleepiness was common (mean ESS 8.5 ± 4.4), and 28% of athletes reported clinically significant sleepiness (ESS ≥ 10). Self-reported snoring (38%) and witnessed apnea episodes (8%) indicated the presence of OSA-related symptoms in this population.	Sleep quality varied across the competitive season, with poorer sleep reported during the pre-season compared with the post-season. Athletes training before 8:00 a.m. exhibited shorter sleep duration and higher levels of daytime sleepiness than those training later in the day. Younger athletes (<20 years) reported higher ESS scores. Rugby union players more frequently reported OSA-related symptoms compared with athletes from other team sports.

OSA—Obstructive Sleep Apnea, SDB—Sleep-Disordered Breathing, ESS—Epworth Sleepiness Scale, PSQI—Pittsburgh Sleep Quality Index, MAP—Multivariable Apnea Prediction Index, MAD—Mandibular Advancement Device, REI—Respiratory Event Index.

## Data Availability

The data are available upon request.

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
