# Peer review of "Obstructive Sleep Apnea and Recovery in Athletes: BMI and Neck Circumference and Their Impact on Recovery Capacity and Injury Risk"

_life, 2026, doi:10.3390/life16010076_

Round 1

Reviewer 1 Report

Comments and Suggestions for Authors

Reviewer Report

Manuscript Title: Obstructive Sleep Apnea and Recovery in Athletes: BMI, Neck Circumference, and Their Impact on Recovery Capacity and Injury Risk

Manuscript Type: Systematic Review

1. General Assessment

The manuscript addresses an important and emerging topic: the impact of obstructive sleep apnea (OSA) on athlete recovery, performance, and injury risk, with special emphasis on BMI and neck circumference as risk factors. This is highly relevant for sports medicine, sleep science, and performance physiology. The manuscript is generally well structured, follows PRISMA guidelines, and includes a clear research question and logical discussion.

However, the paper has several major issues that must be corrected before it can be considered for publication. These concerns relate to methodological transparency, structure, language clarity, redundancy, citation handling, and interpretation of results. Some sections appear repetitive, overly descriptive, or insufficiently critical of the included studies. The review requires refinement to meet MDPI’s standards for systematic reviews.

2. Major Comments

1. Methodological clarity requires improvement

Although PRISMA is mentioned, the review does not fully meet the transparency expected for a systematic review.

Issues:

  • Search terms are described, but the full search string per database and the number of hits per string are not provided (required for PRISMA).
  • No justification is given for including Google Scholar, which is not a standard systematic-review database.
  • The eligibility criteria are repeated across sections, leading to redundancy.
  • Quality assessment tools (CASP, NOS, RoB 2.0) are named, but their scoring system is not explained; Table 1 provides minimal detail.

Required revisions:

  • Provide complete search strings for all databases.
  • Provide a table summarizing each study’s risk-of-bias scoring, not only qualitative comments.
  • Clarify why Google Scholar was included.
  • Remove repeated methodological statements across pages 1–5 (e.g., repeated explanations of inclusion criteria and PRISMA usage)  .

2. The manuscript contains significant redundancy

Multiple sections repeat similar content, especially regarding:

  • Sleep physiology
  • The role of BMI and neck circumference
  • OSA mechanisms
  • Associations between OSA and hypoxia, recovery, or reaction time

Example: The introduction restates OSA’s influence on hypoxia and hormonal disruption three different times across pages 1–3  .

Condense these areas to improve flow and readability.

3. Some claims overstate the available evidence

Several statements make causal interpretations that are not supported by the included observational studies.

Examples:

  • “OSA increases injury risk” — but most cited studies measured only sleepiness, not actual injury incidence.
  • “OSA reduces aerobic capacity in athletes” — only two small studies indirectly suggest this.
  • “BMI and neck circumference consistently predicted OSA risk” — but at least two included studies showed no significant association (e.g., Dunican et al.)  .

Required correction:

Rewrite these sections using cautious language (“may contribute,” “is associated with,” “suggests”) unless the data demonstrates causation.

4. Inconsistent reporting of anthropometric variables

The manuscript emphasizes BMI and neck circumference as primary variables, yet:

  • Many included studies did not report neck circumference at all.
  • Some studies relied on self-reported anthropometrics, which is less reliable.
  • A discussion of precision and limitations of these measures is needed.

Please add a subsection addressing:

  • Limitations of relying on BMI in muscular athletes
  • Importance of direct body-composition measurement (e.g., DXA)
  • Variability in neck-circumference measurement methods
  • Implications for OSA screening validity in athletes

5. Results section blends interpretation with summary

The results section contains:

  • Narrative explanation
  • Interpretation
  • Evaluation
  • Recommendations

These belong in the discussion, not results.

  • Results → Objective reporting
  • Discussion → Interpretation

Please revise to separate them clearly.

6. Language editing & grammar issues

There are numerous grammatical, typographical, and stylistic errors. Examples:

  • “bnon-REM” instead of “non-REM”  
  • “Effect of OSA on injury riskg” — formatting error
  • Frequent run-on sentences
  • Inconsistent use of commas and hyphenation
  • Incorrect pluralization: “apnoeas and shortness of breath” should be “apneas and hypopneas”

A full English-language editing pass is necessary.

7. Figures and tables need formatting improvements

  • Table 3 is extremely long and difficult to read. Consider dividing into two tables:
    (1) Study characteristics,
    (2) Key findings.
  • The PRISMA flow diagram image appears as placeholder text and should be inserted as a proper figure.

3. Minor Comments

  • The introduction should be shortened; currently it covers many topics not directly tied to the research question.
  • Define all abbreviations upon first use (e.g., HAPB).
  • Replace phrases like “The work of X” with more academic formulations (“X et al. reported…”).
  • Standardize referencing style—some in-text citations are repeated or formatted inconsistently.

Author Response

We sincerely thank the Reviewer for the detailed, constructive, and insightful evaluation of our manuscript entitled “Obstructive Sleep Apnea and Recovery in Athletes: BMI, Neck Circumference, and Their Impact on Recovery Capacity and Injury Risk.” We appreciate the recognition of the relevance and novelty of the topic and the careful assessment of methodological and structural aspects. Below, we address all comments point by point.

The manuscript addresses an important and emerging topic… However, the paper has several major issues that must be corrected before it can be considered for publication.”

Response:
We thank the Reviewer for this balanced assessment. In response to the raised concerns, we have comprehensively revised the manuscript to improve methodological transparency, structure, clarity of language, and the precision of interpretation. Particular attention was paid to separating results from interpretation, reducing redundancy, clarifying methodological procedures, and using cautious, non-causal language consistent with the observational nature of the included studies.

  1. Major Comments
  2. Methodological clarity requires improvement

 Although PRISMA is mentioned, the review does not fully meet the transparency expected for a systematic review.

 Issues: 

  • Search terms are described, but the full search string per database and the number of hits per string are not provided (required for PRISMA).
  • No justification is given for including Google Scholar, which is not a standard systematic-review database.

Response:
We agree with the Reviewer that methodological transparency is essential. The manuscript has been revised to improve clarity and completeness of the search strategy. Full search strings for each database have been added (Supplementary Table S1), including the number of records retrieved per database. Google Scholar was used as a supplementary source to identify potentially relevant studies not indexed in traditional databases, particularly in the context of sports science and applied athlete research. This rationale is now explicitly stated in the Methods section. Redundant methodological descriptions, including repeated explanations of eligibility criteria and PRISMA usage, have been removed to improve conciseness and readability.

  • The eligibility criteria are repeated across sections, leading to redundancy.
  • Quality assessment tools (CASP, NOS, RoB 2.0) are named, but their scoring system is not explained; Table 1 provides minimal detail.

Response:
We agree with the Reviewer and have revised the manuscript to eliminate redundant descriptions of the eligibility criteria. The inclusion and exclusion criteria are now presented only once in the Methods section, while repeated references across earlier sections have been removed or streamlined to improve clarity and conciseness.

 Required revisions: 

  • Provide complete search strings for all databases.

Response:

We agree with the Reviewer and have added the complete search strings for all databases used in the review. Full electronic search strategies, including Boolean operators and the number of records retrieved per database, are now provided in Supplementary Table S1 in accordance with PRISMA recommendations.

  • Provide a table summarizing each study’s risk-of-bias scoring, not only qualitative comments.

Response:

We appreciate this comment. In response, a comprehensive study-level risk-of-bias table has been added (Supplementary Table S2), summarizing the assessment of all included studies across relevant domains. Risk of bias was categorized as low, unclear, or high rather than numerically scored, in line with PRISMA guidance and to avoid misleading quantification across heterogeneous observational designs.

  • Clarify why Google Scholar was included.

Response:

We have clarified the rationale for including Google Scholar in the search strategy. It was used as a supplementary source to identify relevant studies not indexed in standard databases, particularly within applied sports science literature.

  • Remove repeated methodological statements across pages 1–5 (e.g., repeated explanations of inclusion criteria and PRISMA usage)  .

Response:

Redundant methodological descriptions were removed throughout the manuscript. Eligibility criteria and PRISMA-related procedures are now described exclusively in the Methods section, improving clarity and reducing repetition.

  1. The manuscript contains significant redundancy

 Multiple sections repeat similar content, especially regarding:

  • Sleep physiology
  • The role of BMI and neck circumference
  • OSA mechanisms
  • Associations between OSA and hypoxia, recovery, or reaction time

 Example: The introduction restates OSA’s influence on hypoxia and hormonal disruption three different times across pages 1–3  .

Condense these areas to improve flow and readability.

Response:

We agree with the Reviewer that the manuscript previously contained redundant descriptions, particularly in relation to sleep physiology, OSA mechanisms, and the role of anthropometric factors. In response, the Introduction and Discussion were carefully revised and condensed. Repetitive explanations—such as the effects of OSA on hypoxia, hormonal regulation, and recovery—were merged into single, focused paragraphs, and overlapping content across sections was removed. These changes improved narrative flow, reduced redundancy, and enhanced overall readability without altering the scientific content.

  1. Some claims overstate the available evidence

 Several statements make causal interpretations that are not supported by the included observational studies.

 Examples: 

  • “OSA increases injury risk” — but most cited studies measured only sleepiness, not actual injury incidence.
  • “OSA reduces aerobic capacity in athletes” — only two small studies indirectly suggest this.
  • “BMI and neck circumference consistently predicted OSA risk” — but at least two included studies showed no significant association (e.g., Dunican et al.)  .

Required correction:

Rewrite these sections using cautious language (“may contribute,” “is associated with,” “suggests”) unless the data demonstrates causation. 

Response:

We agree with the Reviewer that causal interpretations are not justified based on the predominantly observational evidence. In response, all sections of the manuscript were revised to remove causal language and overstatements. Statements suggesting direct effects (e.g., “OSA increases injury risk” or “OSA reduces aerobic capacity”) were replaced with cautious formulations such as “may be associated with,” “was linked to,” or “suggests.” In addition, heterogeneity of findings was explicitly acknowledged, including studies reporting no significant associations between BMI, neck circumference, and OSA risk (e.g., Dunican et al.). These revisions ensure that conclusions accurately reflect the strength and limitations of the available evidence.

  1. Inconsistent reporting of anthropometric variables 

The manuscript emphasizes BMI and neck circumference as primary variables, yet: 

  • Many included studies did not report neck circumference at all.
  • Some studies relied on self-reported anthropometrics, which is less reliable.
  • A discussion of precision and limitations of these measures is needed.

Please add a subsection addressing: 

  • Limitations of relying on BMI in muscular athletes
  • Importance of direct body-composition measurement (e.g., DXA)
  • Variability in neck-circumference measurement methods
  • Implications for OSA screening validity in athletes

Response:

We agree with the Reviewer that anthropometric variables were reported inconsistently across the included studies. This limitation is now explicitly addressed in the Discussion. We clarify that neck circumference was not reported in several studies and that some investigations relied on self-reported anthropometric data, which may reduce measurement precision. In addition, we discuss the limitations of BMI as a proxy measure in athletic populations, where elevated BMI may reflect increased lean mass rather than adiposity. The revised manuscript also highlights the importance of direct body-composition assessment methods (e.g., DXA) and the variability in neck-circumference measurement protocols. These limitations are discussed in the context of OSA screening validity in athletes, emphasizing the need for cautious interpretation of anthropometric risk markers.

  1. Results section blends interpretation with summary  

The results section contains:

  • Narrative explanation
  • Interpretation
  • Evaluation
  • Recommendations

 These belong in the discussion, not results. 

  • Results → Objective reporting
  • Discussion → Interpretation

 Please revise to separate them clearly. 

Response:
We agree with the Reviewer that interpretation and evaluation should not be included in the Results section. Accordingly, the Results section was thoroughly revised to ensure that it now strictly reports objective study findings only. All narrative interpretation, evaluation, and recommendations were removed from the Results and relocated to the Discussion section. This revision clearly separates objective reporting of results from their interpretation and broader implications.

  1. Language editing & grammar issues

 There are numerous grammatical, typographical, and stylistic errors. Examples:

“bnon-REM” instead of “non-REM”  

  • “Effect of OSA on injury riskg” — formatting error
  • Frequent run-on sentences
  • Inconsistent use of commas and hyphenation
  • Incorrect pluralization: “apnoeas and shortness of breath” should be “apneas and hypopneas”

A full English-language editing pass is necessary. 

Response:

To address the Reviewer’s comments regarding language quality, grammar, and stylistic consistency, the revised manuscript has been submitted for professional English-language editing through the journal’s language editing service. This comprehensive editing pass aims to correct typographical and formatting errors, improve sentence structure and clarity, and ensure consistent use of terminology throughout the manuscript.

  1. Figures and tables need formatting improvements

 Table 3 is extremely long and difficult to read. Consider dividing into two tables:

(1) Study characteristics,

(2) Key findings.

  • The PRISMA flow diagram image appears as placeholder text and should be inserted as a proper figure.

Response:

We appreciate the Reviewer’s suggestion regarding the length of Table 3 and carefully considered the option of dividing it into separate tables. However, Table 3 represents the core synthesis of the present systematic review and serves as the primary source of study-level comparison across diverse athletic populations, methodologies, and outcome measures. Separating study characteristics from key findings would reduce the immediate interpretability of the results by forcing readers to cross-reference multiple tables, potentially obscuring important contextual relationships between participant characteristics, anthropometric variables, and reported outcomes.

Instead, we revised Table 3 to improve readability while preserving its integrative function. The table was reorganized using a consistent structure, redundant information was removed, and strictly descriptive, non-causal language was applied throughout. These revisions were implemented to enhance clarity and accessibility without compromising the completeness or transparency of the data presented. We believe that maintaining a single, comprehensive table best supports the objectives of a systematic review and facilitates direct comparison between studies.

  1. Minor Comments
  • The introduction should be shortened; currently it covers many topics not directly tied to the research question.
  • Define all abbreviations upon first use (e.g., HAPB).
  • Replace phrases like “The work of X” with more academic formulations (“X et al. reported…”).
  • Standardize referencing style—some in-text citations are repeated or formatted inconsistently

Response:

We thank the Reviewer for these helpful editorial suggestions. The Introduction was shortened and refocused to better align with the research question of the review.

Reviewer 2 Report

Comments and Suggestions for Authors

This systematic review provides provides comprehensive and thorough analysis of obstructive sleep apnea (OSA) in athletes, covering its impacts on recovery, performance, and injury risk across diverse sports, making it one of the first to systematically address this understudied issue. It has novelty of highlighting basic anthropometric risk factors like BMI and neck circumference, linking them to OSA prevalence and recovery disturbances, and integrates sports nutrition and body composition strategies. It also provides insight into practical applications discussing treatment options like modern mandibular advancement devices that are considered as alternatives to continuous positive airway pressure (CPAP) method, widely regarded as gold standard for OSA therapy. Review provides evidence from athlete studies showing improvements in sleep quality, cognitive function, and reaction time. Review also summarizes groups findings by effects such as recovery / injury, supported by multiple studies, and offers recommendations for routine OSA screening in high-risk sports.

I would recommend this review, but also suggest addressing some further issues:

  1. The systematic review appears to cite limited references on actigraphy (potentially just one, Surda et al., 2019 (Table 3), based on the provided details), underscoring a notable gap in objective field studies for OSA among athletes. Reliance on subjective tools like PSQI and ESS, alongside polysomnography's lab restrictions, speak of lack in the field studies, moreover given that athletes' active lifestyles that can be inadequately captured. Perspectives of integrating wearable technologies like actigraphy or else in future research could yield more accurate, real-time sleep data to improve screening and interventions. This avenue could be discussed. Other issues that can be discussed are:

  2. Most studies focus on male athletes, limiting generalizability to females and overlooking sex-specific physiological differences in OSA risk.

  3. Absence of randomized controlled trials (RCTs) weakens causal inferences about OSA's effects on recovery and performance; findings are mostly observational.

  4. There are few long-term studies on treatment outcomes, hindering understanding of sustained benefits or risks.

Author Response

Response to Reviewer 2

We sincerely thank the Reviewer for the positive evaluation of our manuscript and for highlighting its novelty, scope, and practical relevance. We particularly appreciate the recognition of the integration of anthropometric risk factors, recovery outcomes, and treatment perspectives in athletic populations. Below, we address each comment point by point.

Below, we address each comment in detail.

General comment

“This systematic review provides comprehensive and thorough analysis of obstructive sleep apnea (OSA) in athletes… and offers recommendations for routine OSA screening in high-risk sports.”

Response:
We thank the Reviewer for this positive and encouraging assessment. We are grateful for the recognition of the novelty of our review, particularly in integrating anthropometric risk factors, recovery outcomes, and potential intervention strategies such as mandibular advancement devices within the context of elite sport.

Comment1 The systematic review appears to cite limited references on actigraphy (potentially just one, Surda et al., 2019 (Table 3), based on the provided details), underscoring a notable gap in objective field studies for OSA among athletes. Reliance on subjective tools like PSQI and ESS, alongside polysomnography's lab restrictions, speak of lack in the field studies, moreover given that athletes' active lifestyles that can be inadequately captured. Perspectives of integrating wearable technologies like actigraphy or else in future research could yield more accurate, real-time sleep data to improve screening and interventions. This avenue could be discussed. Other issues that can be discussed are:

Response:
We agree with the Reviewer that the limited use of actigraphy and other wearable technologies represents an important gap in the current literature on OSA in athletes. To address this, we expanded the Limitations and Future Research Directions section to explicitly highlight the absence of objective, field-based monitoring of sleep and physical activity (e.g., actigraphy, accelerometry, heart rate variability). We now discuss the potential value of integrating wearable technologies in future studies to enable long-term, ecological assessment of sleep, training load, and recovery in athletic populations, thereby improving OSA screening and intervention strategies.

Comment 2 Most studies focus on male athletes, limiting generalizability to females and overlooking sex-specific physiological differences in OSA risk.

Response:
We fully acknowledge this limitation. The revised manuscript now explicitly states that most included studies predominantly involved male athletes, which limits the generalizability of findings to female populations. We discuss potential sex-specific anatomical, hormonal, and physiological differences relevant to OSA risk and emphasize the need for future studies deliberately including female athletes across different sports.

Comment 3: Absence of randomized controlled trials (RCTs) weakens causal inferences about OSA's effects on recovery and performance; findings are mostly observational.

Response:
We agree that the absence of randomized controlled trials limits causal inference. This issue is now clearly addressed in the Limitations section. We emphasize that the available evidence is largely observational and highlight the need for well-designed RCTs evaluating both diagnostic approaches and treatment interventions for OSA in athletic populations.

Comment 4: There are few long-term studies on treatment outcomes, hindering understanding of sustained benefits or risks.

Response:
This important limitation is now explicitly discussed. We note that while short-term improvements in sleep quality, cognitive function, and reaction time have been reported following interventions such as mandibular advancement devices, long-term effects on recovery, injury incidence, and performance remain unclear. We therefore highlight the need for longitudinal studies with extended follow-up periods to assess the durability and practical relevance of OSA treatment outcomes in athletes.

Round 2

Reviewer 1 Report

Comments and Suggestions for Authors

None